



Atmospheric
Chemistry
and Physics

# Airborne and ground-based measurements of aerosol optical depth of freshly emitted anthropogenic plumes in the Athabasca Oil Sands Region

**Konstantin Baibakov[1,a], Samuel LeBlanc[2,3], Keyvan Ranjbar[4], Norman T. O'Neill[4], Mengistu Wolde[1], Jens Redemann[5], Kristina Pistone[2,3], Shao-Meng Li[6,7], John Liggio[6], Katherine Hayden[6], Tak W. Chan[8], Michael J. Wheeler[6], Leonid Nichman[1], Connor Flynn[5], and Roy Johnson[3]**

[1]Flight Research Laboratory, National Research Council Canada, Ottawa, Canada
[2]Bay Area Environmental Research Institute, Moffett Field, CA, USA
[3]NASA Ames Research Center, Moffett Field, CA, USA
[4]Dépt. de géomatique appliquée, Centre d'Applications et de Recherches en Télédétection, Université de Sherbrooke, Sherbrooke, QC, Canada
[5]School of Meteorology, University of Oklahoma, Norman, OK, USA
[6]Air Quality Process Research Section, Environment and Climate Change Canada, Toronto, ON, Canada
[7]College of Environmental Science and Engineering, Peking University, Beijing, 100871 China
[8]Climate Chemistry Measurement and Research, Climate Research Division, Environment and Climate Change Canada, ON, Canada
[a]now at: Canadian Space Agency, Saint-Hubert, Canada

**Correspondence:** Konstantin Baibakov (konstantin.baibakov@canada.ca)

**Abstract.** [TS1] In this work we report the airborne aerosol optical depth (AOD) from measurements within freshly emitted anthropogenic plumes arising from mining and processing operations in the Athabasca Oil Sands Region (AOSR) in the context of ground-based AERONET climatological daily averaged AODs at Fort McMurray (Alberta, Canada). During two flights on 9 and 18 June 2018, the NASA airborne 4STAR (Spectrometers for Sky-Scanning, Sun-Tracking Atmospheric Research) Sun photometer registered high fine-mode (FM, < 1 µm) in-plume AODs of up to 0.4 and 0.9, respectively, in the vicinity of the plume source (< 20 km). Particle composition shows that the plumes were associated with elevated concentrations of sulfates and ammonium. These high AODs significantly exceed climatological averages for June and were not captured by the nearby AERONET instrument (mean daily AODs of $0.10 \pm 0.01$ and $0.07 \pm 0.02$, maximum AOD of 0.12) due possibly to horizontal inhomogeneity of the plumes, plume dilution and winds which in certain cases were carrying the plume away from the ground-based instrument. The average 4STAR out-of-plume (background) AODs deviated only marginally from AERONET daily averaged values. While 4STAR AOD peaks were generally well correlated in time with peaks in the in situ-measured particle concentrations, we show that differences in particle size are the dominant factor in determining the 4STAR-derived AOD. During the two flights of 24 June and 5 July 2018 when plumes likely travelled distances of 60 km or more, the average 4STAR FM AOD increased by 0.01–0.02 over ∼ 50 km of downwind particle evolution, which was supported by the increases in layer AODs calculated from the in situ extinction measurements. Based on these observations as well as the increases in organic mass, we attribute the observed AOD increase, at least in part, to secondary organic aerosol formation. The in-plume and out-of-plume AODs for this second pair of flights, in contrast to clear differences in in situ optical and other measurements, were practically indistinguishable and compared favourably to AERONET within 0.01–0.02 AOD. This means that AERONET was generally suc-

cessful in capturing the background AODs, but missed some of the spatially constrained high-AOD plumes with sources as close as 30–50 km, which is important to note since the AERONET measurements are generally thought to be representative of the regional AOD loading. The fact that industrial plumes can be associated with significantly higher AODs in the vicinity of the emission sources than previously reported from AERONET can potentially have an effect on estimating the AOSR radiative impact.

# 1   Introduction

Atmospheric aerosols play an important role in the Earth's climate. Depending on their properties (such as size, composition and concentration) and climatological factors (such as ambient radiative energy budget and surface albedo), aerosols can either cool the surface by reflecting sunlight back to space or have a net warming effect by directly absorbing this shortwave radiation (e.g. Kaufman et al., 2002; Anderson et al., 2003a; Charlson et al., 2005). Aerosols can also affect cloud properties by modifying droplet size, cloud lifetime and cloud burn-off rates (Ackerman et al., 2000; Wilcox, 2010; McComiskey and Feingold, 2012). Despite recent progress, aerosols, together with clouds, remain the greatest source of uncertainty in climate models (Boucher et al., 2013). This is in part due to the complexity of aerosol-generating processes (natural and anthropogenic), and aerosol transport mechanisms resulting in particles of various size, concentration, chemical composition and mixing state that ultimately determine aerosol radiative impact. A recent work by Matus et al. (2019), based on the multi-sensor A-train observations, estimates that the global mean aerosol direct radiative effect is $-2.40\,\mathrm{W\,m^{-2}}$ with anthropogenic direct radiative forcing accounting for 21 %.

The Athabasca Oil Sands Region (AOSR) in Alberta, Canada, is home to the third-largest oil deposit in the world (behind Venezuela and Saudi Arabia) with an estimated 164.1 billion barrels of recoverable crude oil (as of December 2018; see NRCAN, 2020). In addition to the emissions of anthropogenic pollutant and greenhouse gases (e.g., Charpentier and Bergerson, 2009; Bytnerowicz et al., 2010; Baray et al., 2018; Liggio et al., 2019) the AOSR generates a substantial amount of aerosols, either directly as sulfate, black carbon (BC), primary organic aerosol (POA) and dust (e.g., Landis et al., 2017; Cheng et al., 2018), or from secondary atmospheric processes. In fact, Liggio et al. (2016) showed that the AOSR is a large source of secondary organic aerosol (SOA, formed through gas-to-particle conversion) with formation rates comparable to major metropolitan areas. Understanding the radiative impact of anthropogenically produced aerosols in the AOSR through better characterization of their radiative properties is therefore of significant importance.

The estimation of aerosol radiative impact requires knowledge of key aerosol properties. One of the most important of these properties is the aerosol optical depth (AOD) (e.g., Kahn, 2012), defined as the total vertical column of aerosol extinction (the combined capacity of aerosol particles to scatter and absorb light). The mid-visible AOD (typically reported at wavelengths between 440 and 550 nm) can vary significantly from less than 0.1 for rural sites with background aerosols to more than 3 for aerosol plumes from intense forest fires, dust storms or volcanic eruptions. Model analysis suggests that 95 % of global AOD at 440 nm, however, is below 0.4 (Andrews et al., 2017). AOD measured from or near ground level is commonly retrieved using passive Sun photometry. A Sun photometer measures AOD in several spectral channels and can yield an estimation of particle abundance as well as aerosol size indicators (effective radius ($r_\mathrm{eff}$) of submicron ("fine mode", FM) and supermicron ("coarse mode", CM) modes for example) from the spectral information (O'Neill et al., 2003). There are currently two Cimel CE-318 ground-based Sun photometers operating in the AOSR as part of the global AERONET Sun photometer network: one near Fort McMurray (56.8° N, 111.5° W, operating since 2005) and one near Fort McKay (57.2° N, 111.6° W, operating since 2013).

The analysis of local AOD climatologies, including that of AOSR, serves as a basis for a global aerosol climatology and is essential for validating satellite retrievals (Toledano et al., 2007; Sioris et al., 2017b). Sioris et al. (2017a) analyzed AOD data records for several Canadian AERONET stations, including Fort McMurray, in the context of ground-based particulate matter ($PM_{2.5}$) concentrations and presented annual climatologies for both the FM and CM AODs. They found that Fort McMurray FM AODs (at 500 nm) are generally $< 0.1$ for September–April and $< 0.2$ for May–August. The summer-time AOD increase was attributed to forest fires plumes, and no significant trends in the long-term AOD record were observed. In addition to ground-based measurements, Shinozuka et al. (2011) reported on AOD measurements acquired on board the NASA P-3 aircraft using the AATS-14 Sun photometer (AOD from 14 wavelength bands in the 354–2139 nm range) in the vicinity of Fort McMurray in June and July 2008. While the AATS-14 measured $AOD_{499}$ values that frequently exceeded 1 and at times reached 4, the study was entirely focused on biomass burning aerosols from local forest fires and did not include measurements of the oil sands industrial plumes. Indeed, most airborne studies have predominantly focussed on either naturally produced aerosols (such as forest fires or dust events) and/or significantly aged plumes that have been transported for hundreds or thousands of kilometers (Heald et al., 2006; Chin et al., 2007; Stone et al., 2010). While multiple studies have described cases of extreme pollution in China (e.g., Gu et al., 2018; Qin et al., 2018; Sun et al., 2018) significantly less attention has been given to AOD values from isolated industrialized sources outside of Asia. Given that the

anthropogenic plumes in the AOSR, far from large urban regions, are nearly entirely attributed to a singular industry, the obtained airborne AOD measurements can be used to evaluate the representativeness of AERONET measurements in the context of industrial emissions. This is especially relevant to the AOSR, where secondary aerosol formation is likely to occur away from source areas and associated AERONET sites.

In the spring–summer of 2018, the NASA 4STAR Sun photometer (Spectrometers for Sky-Scanning, Sun-Tracking Atmospheric Research; Dunagan et al., 2013; Shinozuka et al., 2013; Pistone et al., 2019; LeBlanc et al., 2020) was integrated on the National Research Council (NRC) Convair-580 aircraft, as part of a state-of-the-art suite of in situ and remote sensing instruments. The NASA 4STAR acquired airborne hyperspectral AOD data during the Oil Sands Measurement Campaign of 2018 (OSMC) and represented the first airborne Sun photometer deployment to study industrial pollutant plumes from the Alberta oil sands mining and upgrading operations.

In the current work, based on four OSMC flights, we evaluate whether the existing ground-based AERONET measurements capture the full extent of AOD variations in the AOSR by comparing the 4STAR AODs of anthropogenic plumes from mining and processing operations with AERONET climatological and diurnal AODs. In addition, we investigate how the AOD evolves as a function of distance downwind of the plume source, in order to better understand the potential effect of secondary aerosol production, i.e. whether the AOD increases as new particles are being formed – a phenomenon that would be missed by a fixed ground-based instrument located near the source. Finally, we consider how the 4STAR FM and CM AODs relate to other in situ measurements such as particle size, optical absorption and scattering coefficients, and particle chemical composition. The latter analysis provides additional insight into the observed AOD variations and ensures that the 4STAR measurements are physically consistent with other instruments.

## 2 Instrumentation and data processing

### 2.1 Oil Sands Measurement Campaign (OSMC)

The Oil Sands Measurement Campaign 2018 (OSMC2018, hereafter referred to as OSMC) was a follow-up to the OSMC2013 campaign, where the Convair-580 aircraft was deployed to the oil sands region for the first time (see for example Liggio et al., 2016; Li et al., 2017). Figure 1 shows the location of the AOSR facilities overlaid on recent satellite imagery. Two types of flights performed during the OSMC focused on (i) pollutant emissions and (ii) their transformation (i.e., chemical evolution/aging) using "box" (latitude–longitude grid) and "screen" (vertical grid) flight patterns, respectively. Box flights encompassed an entire facility to

characterize pollution emissions from individual oil sands processing facilities, while screen flights were conducted to study the evolution of the pollutants transported away from the source. Each box/screen started at 500 ft a.g.l. ($\sim 152$ m) or more, until reaching the top of the boundary layer typically resulting in 3–6 altitude levels. The transformation flights were conducted as a Lagrangian experiment designed to intercept the same plume at increasing distances from the source perpendicular to the wind direction. The measurements presented here were obtained during the second phase of the campaign between 30 May and 5 July 2018.

### 2.2 Sun photometry measurements

#### 2.2.1 AERONET and CIMEL CE-318

The Cimel CE-318 used at Fort McMurray and Fort McKay are standard ground-based Sun photometers which are part of the global AERONET Sun photometer network (Holben et al., 1998). The CE-318 takes measurements of the direct solar irradiance, at nominal sampling intervals of 3 min in eight spectral bands (340, 380, 440, 500, 675, 870, 1020 and 1670 nm) from which spectral AODs can be calculated. For more information about the AERONET and CIMEL CE318 the reader is referred to Giles et al. (2019). In this work we used AERONET Level 2 (cloud-screened and quality controlled) Version 3 data.

#### 2.2.2 4STAR (Spectrometers for Sky-Scanning Sun-Tracking Atmospheric Research)

The NASA 4STAR instrument combines airborne Sun tracking and sky scanning with diffraction spectroscopy. It is comprised of a movable optical head that protrudes through the top of the aircraft fuselage, an accompanying instrument rack inside the aircraft and optical cables that connect the two (Fig. S1 in the Supplement). The hyperspectral measurements are obtained with two Zeiss spectrometers resulting in a total of 1556 spectral channels between 210 and 1703 nm with sampling resolution of 0.2–1 nm below 1000 and 3–6 nm at longer wavelengths. The nominal calibration accuracy of AOD measurements from 4STAR is dependent on wavelength, time of day, tracking stability and various corrections (such as removal of light absorption by trace gases). This accuracy is typically near 1 % in transmittance, for wavelengths outside gas absorption bands between 355 and 1650 nm, resulting in an AOD uncertainty of 0.01 when the Sun is at zenith. A more detailed technical description of the 4STAR can be found in Dunagan et al. (2013) and Shinozuka et al. (2013).

#### 2.2.3 Spectral Deconvolution Algorithm

We applied the Spectral Deconvolution Algorithm (SDA; O'Neill et al., 2003) in order to separate the total measured AOD into its fine- (submicron) and coarse-mode (su-

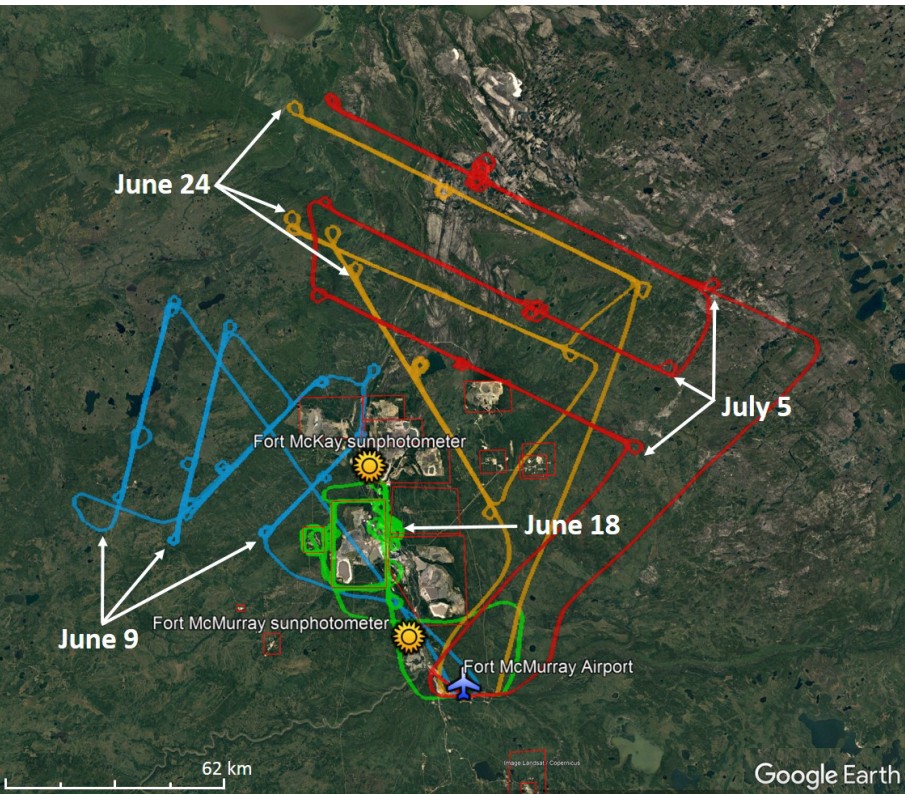

**Figure 1.** Selected flight tracks during the Oil Sands Measurement Campaign. Also shown are locations of AOSR facilities (red outlined boxes) and Fort McMurray and Fort McKay AERONET stations overlaid on satellite imagery (image source: Landsat/Copernicus as used by © Google Earth).

permicron) components ($\tau_f$ and $\tau_c$ or FM and CM). While AERONET uses 380, 440, 500, 675 and 870 nm channels as standard inputs to SDA, using the 380 nm channel in the 4STAR analysis was not possible because of its unrealistically high values (thought to be due to stray light at that wavelength) aggravated by high relative humidity within the spectrometer enclosure. Reducing the number of channels to four is thought to produce results commensurate with the standard AERONET wavelength set with RMS differences in retrieved $\tau_f$ of less than 0.01 (O'Neill et al., 2008). We expect that, in most cases, the anthropogenically produced aerosols observed in the AOSR are submicron, since coarse-mode aerosols are usually associated with natural sources (e.g., winds lifting up dust particles). Consequently, with the exception of dust aerosols resulting from surface mining, the coarse-mode dominant AODs in the AOSR will usually be associated with clouds. As will be shown below, the CM AODs associated with dust in the AOSR are small ($< 0.05$) and can easily be distinguished from cloud-dominated events that are associated with significantly higher AODs and larger variations over short periods of time.

### 2.2.4  4STAR data filtering and cloud screening

In this work we have eliminated all 4STAR measurements with a low (raw count) signal-to-noise ratio and ignored measurements when the aircraft was not in a straight-level track (i.e. when turning or engaged in a spiral). Moreover, in addition to eliminating improbable data (for AOSR) with $AOD_{500} > 3$, we have used the SDA as a spectral cloud-screening where the FM AOD is thought to be free of coarse-mode contamination by either clouds or dust (e.g., O'Neill et al., 2016).

### 2.3  In situ measurements

Ambient air was drawn through a forward-facing, shrouded isokinetic particle inlet (Droplet Measurement Technologies, Boulder, CO, USA). The fine-mode particle size (radius ranging from 30 to 500 nm) was measured using a DMT Ultra-High-Sensitivity Aerosol Spectrometer (UHSAS), which employs light scattering techniques to derive particle concentrations and size distributions (Cai et al., 2008; Kupc et al., 2018). The UHSAS sizing was calibrated using NIST traceable polystyrene latex (PSL) nanospheres. Sizing of the UHSAS is dependent on the refractive index and shape of the particles. Differences in refractive index

have been estimated to result in a 10 % uncertainty in the sizing of the UHSAS (Kupc et al., 2018). For some flights we noticed abnormally high particle counts in five bins between the radius of 0.382 and 0.428 µm. This peak was not supported by other particle spectrometers on the aircraft and is likely an instrument artefact. We removed the problematic data from further analysis and suspect that the issue is related to the uncertainties in the UHSAS calibration curve consisting of several individually chosen gains. The UHSAS effective radius ($r_{\mathrm{eff}}$), defined as an area weighted mean radius of the aerosol particles, was estimated using

$$r_{\mathrm{eff}} = \frac{\int_0^\infty r^3 n(r)\,\mathrm{d}r}{\int_0^\infty r^2 n(r)\,\mathrm{d}r}, \tag{1}$$

where $r$ is the particle radius and $n(r)$ is the particle size distribution (i.e., number of particles per cubic centimeter having a radius in the range $r$ and $r+\mathrm{d}r$ µm). The particle in situ scattering and absorption coefficients were measured using the TSI 3563 nephelometer and Continuous Light Absorption Photometer (CLAP), respectively. The TSI 3563 measures the light scattered by aerosols ultimately yielding the aerosol total scattering coefficient at 450, 550 and 700 nm (Anderson et al., 1996; Bodhaine et al., 1991) while CLAP, developed by the National Oceanic and Atmospheric Administration (NOAA), measures light absorption of particles deposited on 47 mm diameter, glass-fiber filters at 467, 529 and 653 nm (Ogren et al., 2017, and reference therein). Aerosol single scattering albedo (SSA) values were estimated using the ratio of the light scattering coefficient to the sum of the light absorption and scattering coefficients. Black carbon (BC or soot) mass was measured using the DMT Single Particle Soot Photometer (SP2) based on the laser-induced incandescence between approximately 100 and 600 nm (e.g., Schwarz et al., 2006). Finally, non-refractory particle composition (ammonium, nitrate, sulfate and organics) was measured with a High-Resolution Time-of-Flight Aerosol Mass Spectrometer (HR-ToF-AMS; Aerodyne Research Inc., De-Carlo et al., 2006). The AMS was operated in V mode with 10 s time resolution. Ionization efficiency calibrations were performed using monodisperse ammonium nitrate particles prior to, during and after the study with a resulting < 15 % variation. The AMS collection efficiency, ranging from 0.5 to 1.0, was derived by comparing the total AMS mass with the mass derived from the UHSAS size distributions assuming a density consistent with the AMS chemical composition. Background measurements were taken from filtered ambient air approximately 4–5 times per flight. Detection limits, taken from 3 times the standard deviation of the average of filtered time periods, were determined to be 0.122, 0.024, 0.021, and 0.260 µg m$^{-3}$ for NH$_4$, SO$_4$, NO$_3$ and organics, respectively.

## 3 Results and discussion

### 3.1 AERONET AOD climatology

Understanding seasonal AOD variations from the ground-based Fort McMurray instrument is useful for the interpretation of the airborne measurements. While this analysis is required to evaluate whether the AERONET measurements capture the full extent of AOD variations in the AOSR, understanding climatological AODs also provides context for the 4STAR OSMC measurements acquired mostly during a single month of June 2018. Sioris et al. (2017a) discussed some of the Fort McMurray AOD statistics; however, their analysis ends in 2015. We have extended the analysis to 2018 in order to include the period leading up to and coincident with the OSMC.

Figure 2 shows that the total AODs at Fort McMurray are relatively low with values typically less than 0.1 except for summer months (May–August) when monthly averaged AODs can exceed 0.2. There are clear seasonal trends in both the FM and CM AODs. The summer-time FM AOD increase is thought to be predominantly associated with forest fires that frequently occur in the region (Sioris et al., 2017a). For example, the second highest monthly AOD on record of 0.45 occurred during May 2016 when intense forest fires had a major impact on air quality and resulted in the evacuation of more than 80 000 people (Landis et al., 2018). The highest hourly and daily averaged AODs during that month were 2.10 and 1.80, respectively. Figure S2 in the Supplement shows monthly averaged AOD time series for 2005–2018 at Fort McMurray with daily averaged values shown in pink shading. Also, included for reference is a fine-mode AOD time series for nearby Fort McKay; however, the present discussion focuses on Fort McMurray because of its significantly longer measurement record and data availability during OSMC. In many cases, on days with persistently high AOD values (e.g., FM AOD > 0.5) forest fire smoke plumes were clearly discernable in the vicinity of Fort McMurray from satellite imagery (not shown). In comparison, the highest FM monthly AOD average during the non-summer months (September–April) – considered to be primarily impacted by industrial pollution plumes in the absence of other major aerosol sources – was 0.13 with 96 % of all hourly measurements being lower than that value.

With respect to the coarse mode, the peak monthly averaged AOD of 0.04 is associated with the month of April and is thought to be due to dust particles. Landis et al. (2017) demonstrated through systematic chemical analysis of summertime CM aerosol from ground-based measurements that a majority of dust is due to local surface mining activities where heavy machinery (trucks and excavators) lift significant amounts of dust into the air, although long-range transport of Asian dust may be more prominent in the spring (e.g., McKendry et al., 2007; AboEl-Fetouh et al., 2020). AboEl-Fetouh et al. (2020) argued, for example, that a springtime

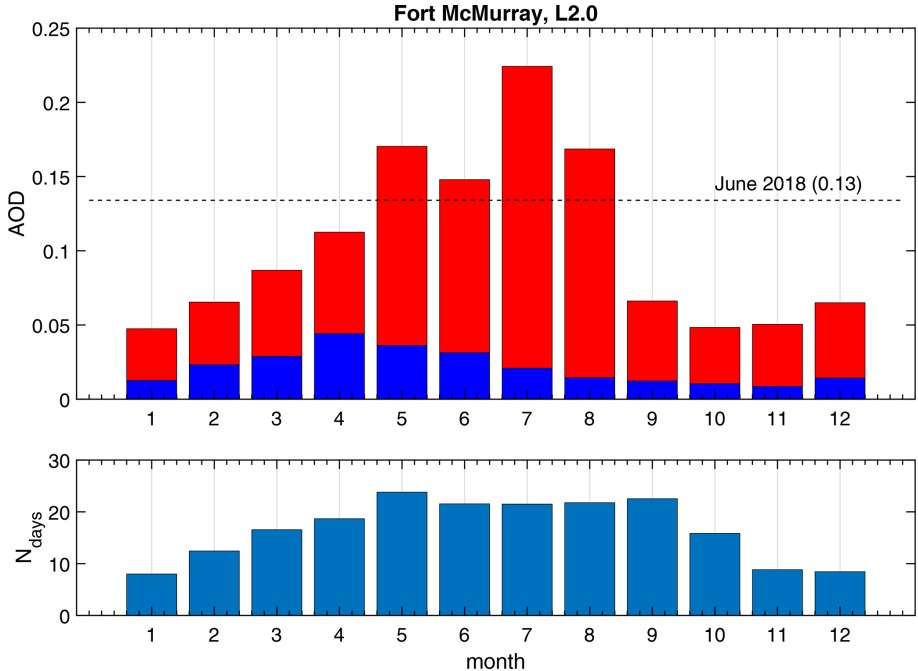

**Figure 2.** Monthly fine- (red) and coarse-mode (blue) AOD averages for the Fort McMurray AERONET station. The dashed line shows the June 2018 mean total AOD. The error bars represent standard deviations of the (total) monthly mean AODs. The bottom graph shows the mean number of days associated with each monthly binned average. These statistics are for AERONET V3, level 2.0, extinction AODs averaged over the data range from 2005 to 2018 inclusively. TS2

coarse-mode event was continental in scale (including the North American and European Arctic) and that it was associated with (relatively small) coarse-mode particles around 1.3 μm in radius.

Figure 2 also indicates that the mean total AOD during June 2018 was $0.13 \pm 0.05$ suggesting that the aerosol loading during the OSMC was generally representative of the climatological AOD values for June (0.15 AOD).

## 3.2 4STAR case studies during OSMC

In this section, based on the OSMC flights of 9 and 18 June, we discuss how the 4STAR AOD variations associated with the AOSR emitted plumes relate to the corresponding aircraft in situ measurements. While direct comparisons between the 4STAR column-based AODs and flight-level in situ aerosol properties are not strictly possible, evaluating the AOD measurements in the context of time-synchronous in situ measurements provides insight into the nature of the plume and the associated AOD variations (e.g., Stone et al., 2010; Shinozuka et al., 2013; Pistone et al., 2019). In particular, understanding how changes in aerosol intensive properties such as particle size and chemical composition might affect the AOD response is necessary to evaluate the radiative importance of the emitted plumes and the associated secondary organic aerosol production.

### 3.2.1 Case study of 9 June 2018

On 9 June, the Convair-580 conducted a transformation flight to the west of the oil sands facilities. The aircraft flew 4 screens perpendicular to the wind direction intercepting the emitted plumes from the majority of the largest surface mining facilities (Fig. 1). Figure 3 shows the AOD time series and corresponding in situ measurements for the relatively cloud-free screen 1, with five nominal altitude levels and "in-plume" and "out-of-plume" periods separated based on elevated values relative to the upwind (background) measurements. While the average fine-mode AODs were $\sim 0.10$ outside of the plume, the values were greater than 0.4 within the plume. Moreover, the two consistent peaks at each altitude level in the UHSAS total particle number concentration (Fig. 3b) suggest that the Convair-580 was intercepting two spatially separated plumes where only one is associated with a significant AOD response. Particle composition data (Figs. 3d and 4) show that the first plume (plume "A", detected by the 4STAR and shaded in pink) contained significantly more $SO_4$ and $NH_4$ than the second plume (plume "B", shaded in grey) indicating that the two plumes likely originated from different facilities. The organic aerosol mass, however, was comparable between the two plumes and similar to the background level. Figures 3c and 5 show a clear distinction in particle effective radius and volume size distribution between plumes A and B in that the latter lacks

optically significant contribution from larger FM particles. The difference is especially pronounced, both in comparison to plume B and out-of-plume (background) values, for particle radii ($r_p$) larger than $\sim 0.12\,\mu m$ with the largest differences observed at radii 0.3 and 0.42 μm TS3. The relative differences between plumes A and B continue to persist for screen two, approximately 20 km downwind (not shown), but the concentration of large FM particles larger than $\sim 0.12\,\mu m$ for plume A was significantly lower (by a factor of 2 or more relative to screen one) as the plume disperses and some of the larger particles settle out of the atmosphere. Based on wind patterns, the plumes on 9 June come mostly from the Syncrude Mildred Lake (SML) and/or Suncor Millenium facilities which perform a number of different processes related to bitumen extraction. The majority of emissions come either from upgrading the bitumen (and in the process emitting $SO_2$ and $SO_4$), mining processes including from big trucks digging and transporting materials, and other unknown processes within the plant itself. The two plumes (A and B) mostly represent emissions and subsequent transformation of pollutants from the upgrading (high $SO_4$ concentrations) but can also contain contributions from mining and tailing ponds. We expect that these two plumes may mix with each other at times.

The 9 June aerosol plume from the oil sands processing could be observed in the MODIS satellite imagery, in between the often-prevalent cloud conditions (see Fig. 6a). The aerosol plume was being advected towards the northwest, with the background aerosol loading in between clouds, mostly with AODs below 0.1. The 4STAR-sampled aerosol plume was observable in the satellite imagery, with $AOD_{500}$ exceeding 0.5. For this case, there are only very limited collocated AOD measurements simultaneously acquired with MODIS and 4STAR. In Fig. 6b all pixels that were measured by MODIS (within 3 km of the flight track) and 4STAR (within 3 h of Terra overpass) are compared. The differences between the 4STAR measurements ($N = 2510$) and the average nearest MODIS pixel show that much of the higher AOD observed by 4STAR (above 0.5) in the middle of the plume is not captured by MODIS (with a maximum AOD of 0.5), and that there is an overestimate at the lower end of the 4STAR-observed AODs. When considering only the nearest 30 min of the MODIS overpass, the cloud-free observations by 4STAR ($N = 327$) are not centered on the optically thickest part of the plume but show MODIS overestimating the AOD by an average of 0.1 (up to a maximum overestimate by 0.25) for the 4STAR AODs at 0.2.

Based on the 4STAR and in situ data, the aerosol plume was $\sim 19$ km wide (spanning plumes A and B) for the first screen (at a distance of emissions of $\sim 20$ km, assuming Syncrude Mildred Lake as a plume source). This evolved to a width of $\sim 31$ km for the last screen, at a further distance of $\sim 70$ km from emissions, although the plume boundaries are harder to identify amid a smaller difference with background values. This is related to the dilution of aerosol over a broader volume, potentially reducing AOD for any one column, but also to a (marginally) larger increase in AOD due to SOA formation (the last screen was heavily contaminated by clouds resulting in practically no useable AOD data). The AOD variations in this plume can be compared to previous observations in the boreal region as reported by Shinozuka and Redemann (2011), which showed that for the Canadian local emissions from forest fires, variations of near 20 % were observed at length scales of 20 to 35.2 km. The aerosol from long-range transport showed variation of less than 5 % (Shinozuka and Redemann, 2011). For the OSMC plumes observed here, a relative change in AOD of about 20 % was observed at distances on a distance scale of 50 km, but only for a subset of the flights (5 July and 24 and 18 June, example of which showing differences between AERONET and 4STAR AOD as a function of distance; Fig. S5 in the Supplement). The differences in length scales of the aerosol plumes are indicative of both the industrial processes and the meteorological conditions and seem to be different than the plumes from biomass burning in the boreal forest.

### 3.2.2 Case study of 18 June 2018

During the 18 June emissions flight around the SML facility, there was a significant and clearly visible industrial plume emanating from the SML stacks (Fig. S3 in the Supplement). Figure 7 shows the corresponding data for the first three altitude levels: 500, 750 and 1000 ft ($\sim 152$, $\sim 229$ and $\sim 305$ m) with the 500 ft track repeated twice. The 4STAR measurements show that for the duration of the Syncrude box the total AOD at 500 nm was relatively low and stable around 0.05–0.07, with AOD spikes of up to 0.86 associated with the plume on the eastern side of the box. As in the case of 9 June, the measurements show that the plume was heterogeneous with two distinct regions identified in Fig. 7 in pink (plume "A") and grey ("plume "B"). Plume A is characterized by a significant response in fine-mode AOD and increased (up to a factor of 2) UHSAS effective radius. Plume A is also associated with high values of $SSA_{440}$ (up to 0.96, Fig. 7d), indicating that the particles were mostly of a scattering nature. In contrast, the SSA values dip below 0.75 in plume B suggesting a much stronger presence of absorbing particles. This hypothesis is confirmed by the enhanced concentrations of highly absorptive BC particles (Fig. 7e), with BC mass increasing during certain times by more than a factor of 10 relative to plume A. Similarly to 9 June, plume A contains significantly higher concentrations of $SO_4$ and $NH_4$ than plume B (not shown). It should be noted that the AOD peaks precede most of the plume A in situ peaks, which we attribute to the 4STAR viewing geometry in relation to the vertical extent of the plume (i.e. the relative position of the Sun, the plume and the aircraft carrying the 4STAR). Plume B contains fewer of the larger FM particles and goes mostly undetected by 4STAR. At the same time, plume B contains higher concentrations of CM particles from surface mining

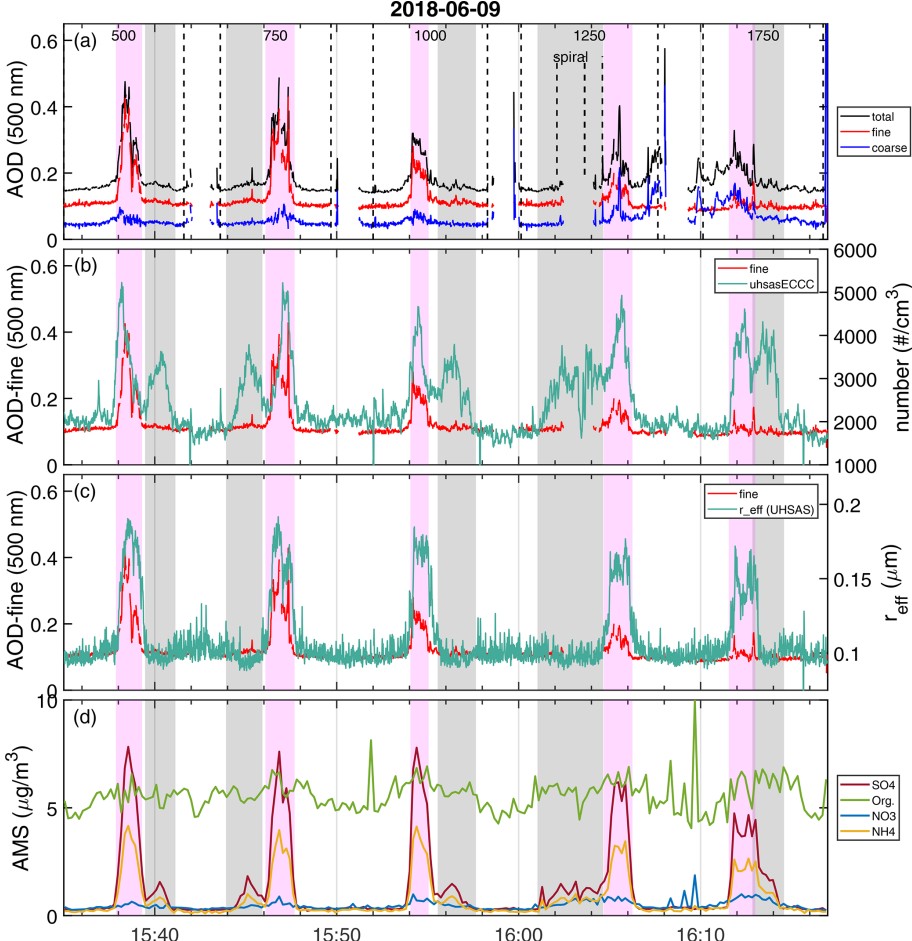

**Figure 3.** Screen 1 time series on 9 June (UTC). **(a)** total, fine-mode and coarse-mode AODs at 500 nm derived from the SDA of the 4STAR data; **(b)** UHSAS total particle number concentration (all sizes), number per cubic centimeter; **(c)** UHSAS effective radius, µm; **(d)** AMS particle composition. In **(a)**, the dashed vertical lines delineate each altitude level with nominal (planned) altitudes listed in ft a.g.l. at the top. The fine-mode AOD is reproduced on several plots to facilitate the comparisons. The pink and grey rectangles represent periods associated with plumes "A" and "B", respectively, while the green shading indicates periods when the aircraft was engaged in a spiral manoeuvre. TS4

(not shown), but the resulting CM AOD (for example, around 15:22) is < 0.03, which is significantly smaller than the FM AODs (all times in this paper refer to UTC).

For both flights, 9 and 18 June, the analysis indicates that differences in particle size within plumes are the driving factor responsible for the AOD response. As a consequence, while certain sections of the plume (e.g., plume B) are clearly associated with elevated in situ particle and gas concentrations, they have little effect on AOD when particulates are small (in this case $r_p < 120$ nm).

### 3.3 4STAR-AERONET comparisons

The inherent assumption of using the AERONET data is that it is representative of the regional AOD within a certain distance from the ground-based station. While this distance is often taken as 100 km, the spatial representation will ultimately depend on the local context including proximity to

aerosol sources and meteorology (Anderson et al., 2003b; Holben et al., 2018). Consequently, in this section we investigate the representativeness of Fort McMurray AERONET measurements of AOSR AODs by studying how the in-flight AOD variations compare to climatological and diurnal ground-based AODs.

Figure 8 and Table 1 show AOD vertical profiles and AOD statistics for three transformation flights (9 and 24 June and 5 July) and one emission flight (18 June) during the OSMC. For 9 June we only included data from screen 1 and 2, beyond which the measurements were judged to be contaminated by clouds. The in-plume data were selected based on significant increases in the 4STAR, UHSAS, nephelometer and CLAP signals relative to the background (out of plume) levels. The examples of the defined "in-plume" time periods are shown in Figs. 3 and 7. While each screen was usually performed at fixed nominal altitudes (e.g., 500, 750 ft a.g.l.), the terrain variations result in slightly different altitudes above sea level

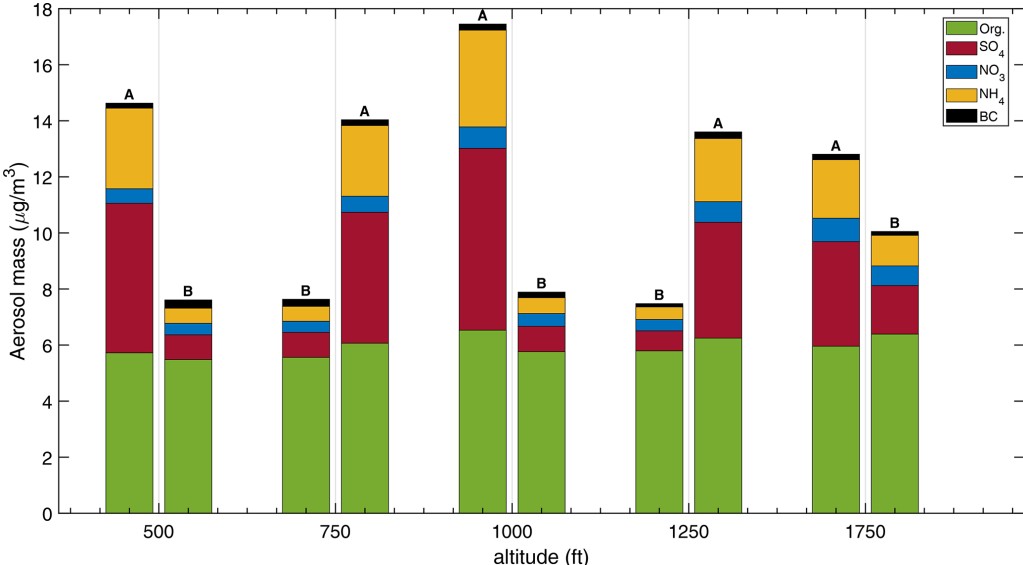

**Figure 4.** Particle chemical composition for plumes A and B during screen 1 of the 9 June transformation flight.

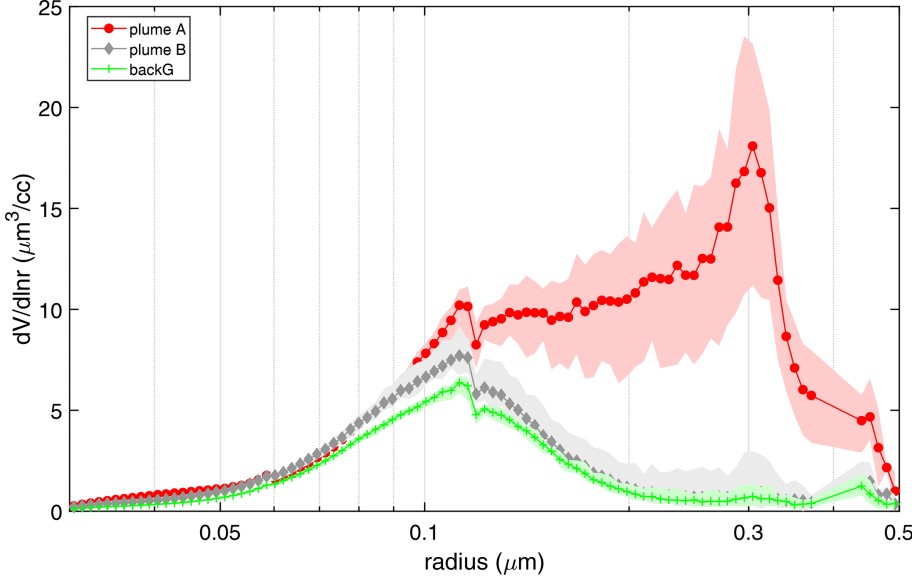

**Figure 5.** Averaged UHSAS volume PSDs during screen 1 on 9 June. Colour patches represent variations with altitude while solid lines show the screen-averaged PSDs for plumes A and B as well as the background (out-of-plume) signal.

when compared screen to screen. Figure 8b also shows fine-mode AERONET daily means and standard deviations for the corresponding dates. It should be noted that on 24 June the AERONET data were significantly affected by the incoming forest fires plume with AODs > 0.4 in the later part of the day. Removing those data after 18:13 ($\sim 27\%$ of points) changes the daily average for 24 June from $0.13 \pm 0.08$ to $0.10 \pm 0.03$.

On 9 (screen 1) and 18 June ("high-AOD flights") the plumes were sampled in the direct vicinity of the source. Consequently, the 9 and 18 June cases were associated with particularly high AODs and show relatively large variations in terms of the vertical distribution and AOD variability within the plume. These large AODs and corresponding variations significantly exceed climatological AODs for June and are clearly not captured by AERONET. However, the average 4STAR out-of-plume (background) AODs deviate only marginally from AERONET daily averaged values (Table 1). During the 24 June and 5 July flights ("low-AOD flights"), the plumes in general travelled longer distances than during the high-AOD flights before being measured by the Convair-580 and hence had more time to mix and disperse. The in-

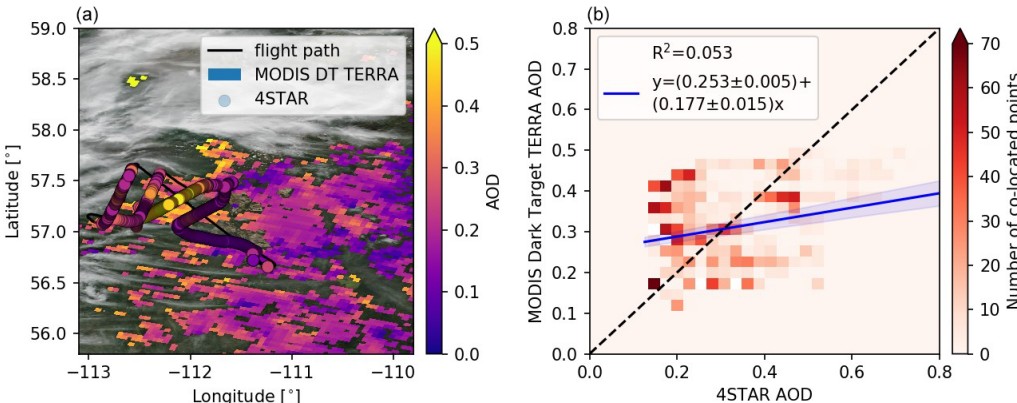

**Figure 6.** Comparison of AOD measured by 4STAR and retrieved by MODIS Dark Target (DT, MOD04_3K; Levy et al., 2015) for 9 June 2018. **(a)** The aerosol scene is overlaid on the true-colour imagery from MODIS TERRA, with the colour bar showing the AOD sampled by MODIS DT (in rectangular pixels) and 4STAR (circles put along the flight track of the NRC Convair-580). **(b)** The pixel-by-pixel comparison of the MODIS DT AOD and 4STAR AOD, with a best fit line in blue, and 95 % confidence intervals of the fit line in blue shading. TS5

**Table 1.** FM AOD statistics for selected flights during the OSMC (mean ± SD). 4STAR in-plume and out-of-plume statistics are reported for the lowest flight level.

| Date (flight no.) | 4STAR in-plume average | 4STAR in-plume maximum | 4STAR out-of-plume average | AERONET daily average | AERONET daily max |
|---|---|---|---|---|---|
| 9 June (F14) | | | | | |
| Screen 1 | 0.24 ± 0.08 | 0.43 | 0.11 ± 0.00 | 0.10 ± 0.01 | 0.12 |
| Screen 2 | 0.16 ± 0.03 | 0.23 | 0.11 ± 0.01 | | |
| 18 June (F19) | 0.18 ± 0.15 | 0.85 | 0.05 ± 0.00 | 0.07 ± 0.02 | 0.12 |
| 24 June (F24) | | | | | |
| Screen 1 | 0.12 ± 0.01 | 0.15 | 0.13 ± 0.01 | 0.13 ± 0.08 | 0.37 |
| Screen 2 | 0.12 ± 0.01 | 0.14 | 0.11 ± 0.01 | 0.10 ± 0.03* | 0.18* |
| Screen 3 | 0.13 ± 0.01 | 0.15 | 0.12 ± 0.00 | | |
| 5 July (F30) | | | | | |
| Screen 1 | 0.04 ± 0.00 | 0.05 | 0.04 ± 0.00 | | |
| Screen 2 | 0.05 ± 0.00 | 0.06 | 0.05 ± 0.01 | 0.05 ± 0.01 | 0.08 |
| Screen 3 | 0.06 ± 0.01 | 0.07 | 0.05 ± 0.01 | | |

* With data from forest fire plumes excluded (∼ 27 % of data).

plume and out-of-plume AODs for these dates were indistinguishable (in contrast to clear differences in in situ measurements, not shown) and the airborne and ground-based averages compared favourably within 0.01–0.02 AOD. In other words, the AERONET measurements capture the background AODs relatively well, but miss some of the spatially constrained high-AOD plumes with sources as close as 30–50 km.

It is not clear what kind of agreement would be considered as reasonable between 4STAR and AERONET given the differences in spatial sampling between the two platforms. While several studies have reported agreements down to RMS differences of 0.01 for wavelengths between 500 and 1020 nm (Stone et al., 2010; Shinozuka et al., 2013; LeBlanc et al., 2020), those comparisons often involve aged plumes that are likely to be more (horizontally) homogeneous than newly emitted AOSR plumes. Furthermore, the comparisons usually include only a few points closest in time to the aircraft overpass of an AERONET station. Factors such as prevalent wind directions, local topography and distance to emission sources can all bias comparisons between the aircraft and ground-based platforms. In particular, during the 9 and 18 June flights, the Fort McMurray AERONET station was mostly located upwind of the Syncrude facilities and

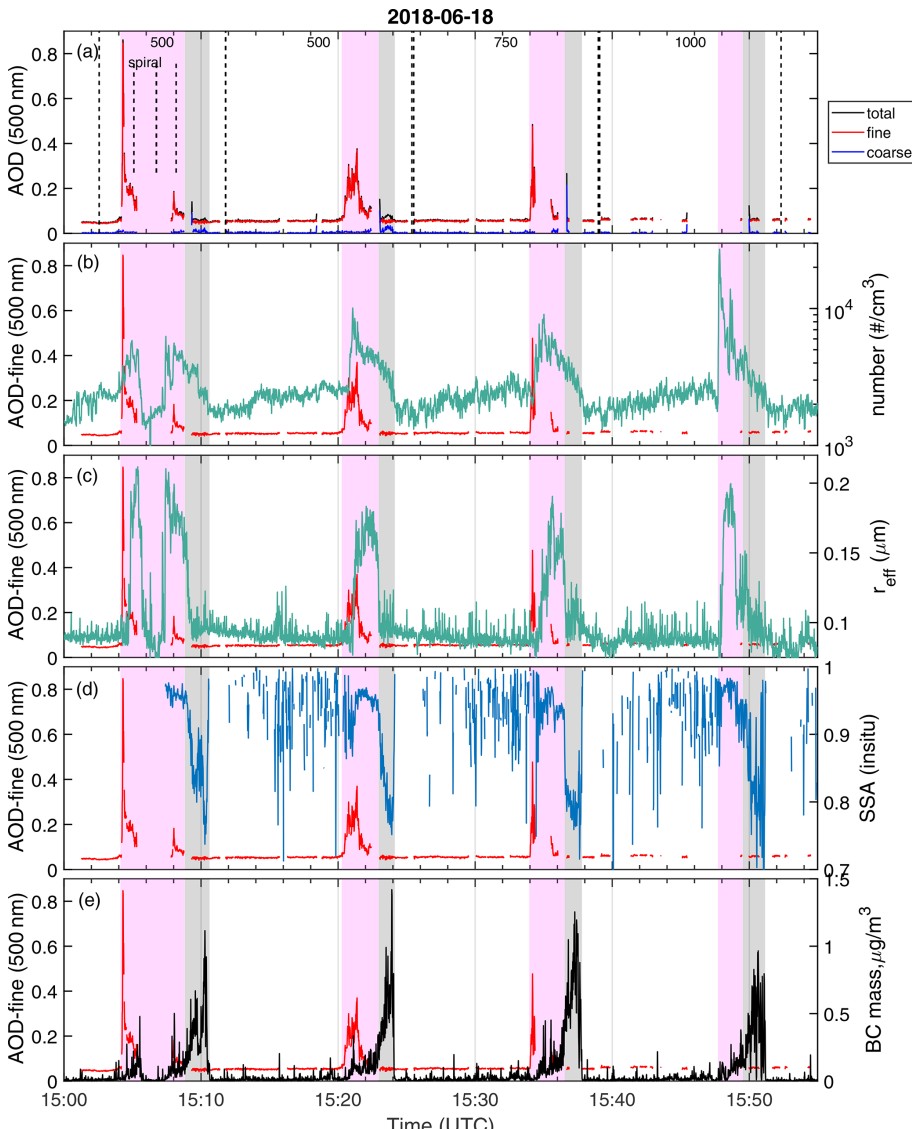

**Figure 7.** Panels **(a)**–**(c)** as in Fig. 3. **(d)** SSA440 from nephelometer and CLAP, unitless; **(e)** black carbon mass, μg m$^{-3}$. The pink and grey rectangles represent plumes A and B respectively as discussed in the text. UTC time.

likely missed most of the plumes measured by the 4STAR. However, even when limiting the AERONET data to periods associated with northern winds (270 to 90° on a wind rose) likely carrying the industrial plumes (Fig. 1), the mean and median June 2018 AODs were both 0.09 with only 4 individual measurements exceeding 0.2 (maximum of 0.29). We attribute these disparities to dilution as the plumes are being transported from their sources to where they are captured by the AERONET Sun photometer. Simply moving the AERONET station closer to specific facilities might yield a better agreement in certain cases, but the AOD differences will likely remain because of spatial inhomogeneity of the plumes and the inherent limitations of measurements from a single fixed ground-based station.

Figure 9a shows 4STAR-AERONET scatterplots for selected flights. Only out-of-plume AODs acquired during a straight-level flight (i.e. excluding spirals) at an altitude of less than 1 km were considered for comparison. Each valid AERONET L2.0 direct-Sun measurement was matched in time with the nearest 4STAR measurement within a 10 min window (the 4STAR-AERONET distance varied between 21 and 154 km), and if no such measurements were found, the AERONET point was excluded from analysis. Even though the average out-of-plume AODs compared favourably to AERONET, the actual linear correlation coefficient ($R^2$) values were relatively low ($\sim 0.30$) likely indicating spatial variations in the background aerosol loading and/or uncertainties in defining out-of-plume sections (there may well

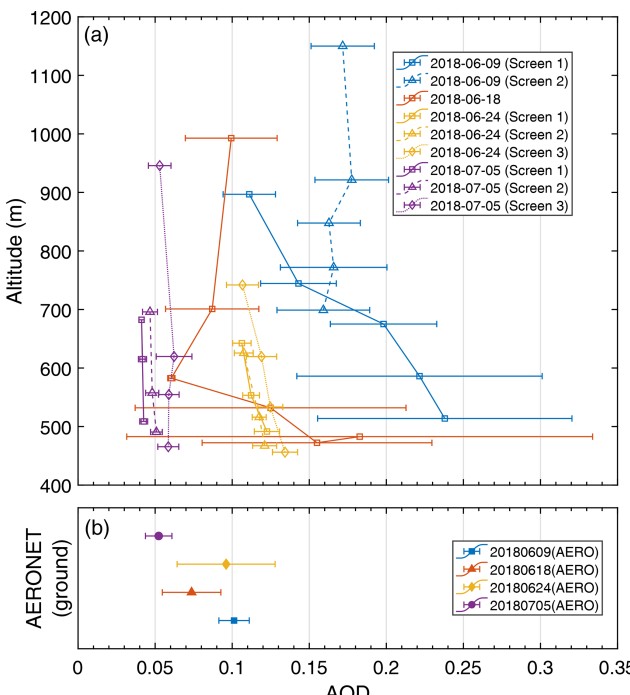

**Figure 8. (a)** 4STAR in-plume mean fine-mode AODs for transformation flights of 9 and 24 June and 5 July and an emission flight of 18 June. **(b)** Fort McMurray ground-based AOD daily averages and standard deviations for selected dates (staggered vertically to show the error bars). In **(a, b)** the horizontal error bars indicate the AOD standard deviation representing spatial variability of the plumes (per altitude SD values for 4STAR and daily SD values for AERONET). For 24 June the AERONET data after 18:13 was removed because it was associated with forest fires.

have been moderately higher AODs in the neighbourhood of those plumes that escaped our plume rejection criteria). A notable exception ($R^2 = 0.65$) was 18 June when both 4STAR and AERONET showed a slight low-frequency AOD 5 increase (from 0.05 to 0.06) during the flight time (Fig. S4 in the Supplement, top panel). The bottom panel of that figure also shows that the AERONET $r_{\mathrm{eff,f}}$ values agree particularly well with UHSAS, but the 4STAR retrievals seem to be overestimated. We suspect that the source of this latter inconsis-10 tency is due to the artificially high 4STAR irradiance signals at the shorter wavelengths. Figure 9b shows a histogram of $\tau_{\mathrm{diff}}$ values (defined as $\tau_{\mathrm{diff}} = \tau_{\mathrm{f,4STAR}} - \tau_{\mathrm{f,AERONET}}$) for all four flights combined. It is evident from this figure that despite the slightly shorter atmospheric column, 4STAR AODs 15 are marginally overestimated relative to AERONET with $|\tau_{\mathrm{diff}}|$ mean, median and standard deviation being 0.02, 0.01 and 0.02. The interpretation of this result is not obvious. In principle, as the aircraft moves further away from sources the AOD should increasingly resemble the background, and 20 thus $\tau_{\mathrm{diff}}$ should decrease with distance. On the other hand, as the distance between the two instruments increases so do the differences in sampled air masses. In fact, Sioris et al.

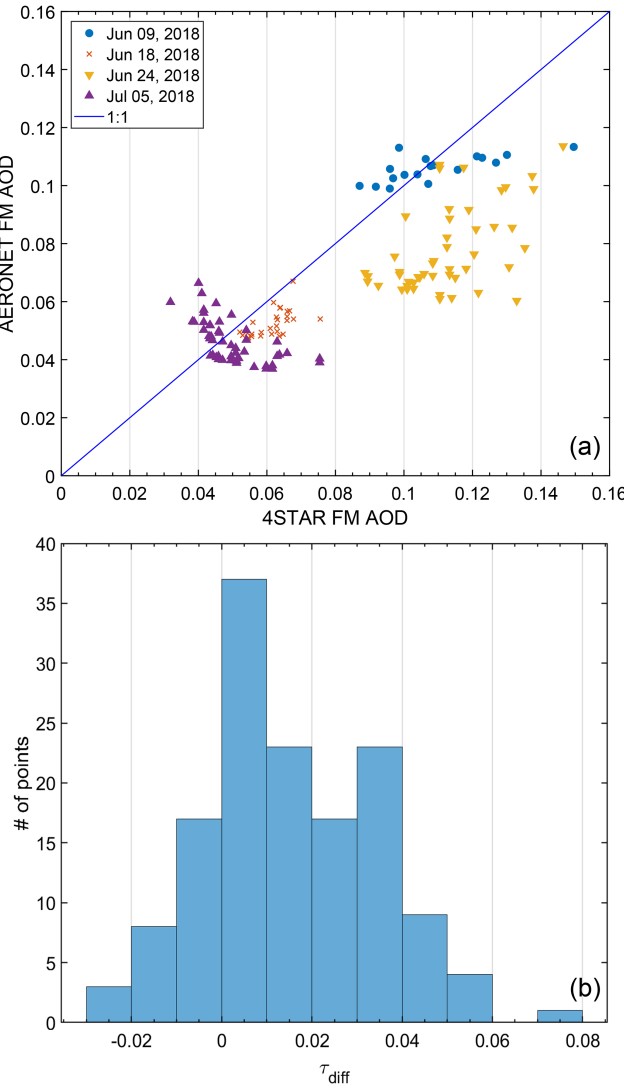

**Figure 9. (a)** AOD fine-mode comparisons between 4STAR and AERONET for the selected flights of 9, 18 and 24 June and 5 July. **(b)** Corresponding histogram of fine-mode AOD differences between the 4STAR and AERONET. AERONET data were matched with 4STAR within 10 min. 4STAR data include only out-of-plume AODs acquired during a straight-level flight (i.e. excluding spirals) at an altitude of less than 1 km. For 24 June the AERONET data after 18:13 were removed because they were associated with forest fires.

(2017a, their Fig. 4) showed that the two AERONET stations become progressively dissimilar as a function of distance between them with the correlation coefficient dropping 25 by ∼ 50 % in the first 500 km. This is consistent with Fig. S5, which indicates that despite the significant variance in the data, the average $\tau_{\mathrm{diff}}$ increases as a function of distance separating the 4STAR and the Fort McMurray AERONET station (up to 160 km) based on the data from the four flights. This 30 suggests that either the 4STAR was influenced by dispersed plumes with AODs higher than the AOSR background even

at larger distances (i.e. not fully diluted plumes), or that the AERONET AODs were influenced by cleaner air masses.

Another potential explanation accounting for some of the 4STAR overestimation is secondary organic aerosol (SOA) production, which may increase with distance and time (e.g. Liggio et al., 2016). This possibility was investigated by targeting the transformation flights of 24 June and 5 July, in an attempt to observe any systematic screen-to-screen AOD changes that might be due to SOA production – a phenomenon that could be missed by AERONET. As shown in Table 1 between screen 1 to 3 only a modest increase of 0.01 and 0.02 was observed. While these AOD changes are small and commensurate with the 4STAR measurement error, O'Neill et al. (2008) previously showed that changes on this scale can be linked to real underlying changes in aerosol properties. We compared the in-plume vertical profiles obtained during screens 2 and 3 of each of the two flights, excluding the data from screen 1 because on 24 June it was not properly positioned relative to other screens and on 5 July the profile data only covered a short vertical range, which we judged to be inadequate for analysis. We found that for both dates the calculated in situ extinction coefficient at 500 nm was higher at practically each level for screen 3 (Fig. 10). Integrating the extinction coefficient over the common vertical range between the screens yielded (in situ) layer AOD increases between 33 % and 67 %. Comparing these values to 4STAR was not always possible in part because of the missing Sun photometry data, and in part because the AOD did not vary sufficiently (i.e. above the measurement uncertainty of 0.01) over these short vertical ranges of 200–500 m to be reliably measured. However, for the two longest profiles during screen 3 on 5 July spanning 800 m, the in situ layer AOD of $0.013 \pm 0.002$ compared favourably to 4STAR AOD of 0.01 (taking standard deviation within a 50 m altitude bin as a measurement error for extinction and assuming that the errors in altitude measurements are negligible). Therefore, the observed 4STAR AOD increases, while small, are consistent with layer AOD increases calculated from in situ extinction measurements. Particle composition data show that the plumes were dominated by the organic content ($> 78 \%$) with the organic mass increasing from screen to screen despite dilution (Fig. S6 in the Supplement). Using a Top-Down Emissions Rate Retrieval Algorithm (TERRA; Gordon et al., 2015) significant SOA formation rates were determined of $1003 \pm 193$ and $443 \pm 45 \, \mathrm{kg \, h^{-1}}$ for 9 June and 5 July, respectively. Consequently, while acknowledging the limited scope of the two flights and the need for more case studies, we believe that the observed AOD increase can be attributed, at least in part, to the process of SOA formation.

## 4 Summary and conclusions

During the spring–summer 2018 Oil Sands Measurement Campaign, the NASA 4STAR Sun photometer acquired air-borne hyperspectral AOD data in the Athabasca Oil Sands Region. In this work, we report on the airborne AOD measurements of anthropogenic plumes from mining and processing operations in AOSR in the context of AERONET climatological and diurnal AODs at Fort McMurray. The monthly averaged AERONET AOD for June 2018, when most of the 4STAR Sun photometer data were acquired, was $0.13 \pm 0.05$ (mean $\pm$ SD), which was in line with summertime climatological averages, indicating that the observational conditions during OSMC were representative of the typical aerosol loading in the area for that month.

During the two flights of 9 and 18 June (high-AOD flights), while flying into the visually opaque industrial plumes, 4STAR registered high fine-mode in-plume AODs of up to 0.4 and 0.9, respectively, in the vicinity of the plume source ($< 20 \, \mathrm{km}$). These high AODs significantly exceed climatological averages and were clearly not captured by the nearby AERONET instrument, which reported mean daily AODs of $0.10 \pm 0.01$ and $0.07 \pm 0.02$ (mean $\pm$ SD) for those days. At the same time the average 4STAR out-of-plume (background) AODs deviated only marginally from AERONET daily averaged values. We attribute the disparities principally to the horizontal inhomogeneity of the plumes, the distance between the plume source and the AERONET instrument as well as prevalent winds which, in certain cases, were carrying the plume away from the ground-based instrument.

For both flights, the 4STAR AOD peaks were generally well correlated in time with peaks in in situ-measured particle and gas concentrations for the optically active sections of the plume (plume "A", i.e. those sections that were associated with significant increases in in situ extinction and/or AOD), while other sections, clearly associated with elevated in situ concentrations (relative to out-of-plume measurements), produced no significant 4STAR response (plume "B"). We attribute this difference to the presence of larger FM particles within plume A (FM effective radius increasing from 0.10 to 0.18 μm) relative to the out-of-plume values and plume B. Particle composition analysis shows that plume A contained elevated levels (by up to a factor of 6) of sulfates and ammonium while the organic mass remained largely comparable between the two plumes.

During the 24 June and 5 July flights (low-AOD flights), the plumes likely travelled longer distances before being measured by the Convair-580 onboard instruments (the Syncrude plume for example would have travelled $\sim 60 \, \mathrm{km}$, or 3 times the distance than on 9 June, before being first measured) and hence had more time to mix and disperse. The in-plume and out-of-plume 4STAR AODs for these dates were practically indistinguishable (average AODs of $\sim 0.12$ and 0.05 for 24 June and 5 July, respectively), and the airborne and ground-based averages compared favourably to within 0.01–0.02 AOD. Comparing screen 1 to 3 AODs for these flights shows a small increase of 0.01–0.02, which was supported in several cases by the increases in layer AOD cal-

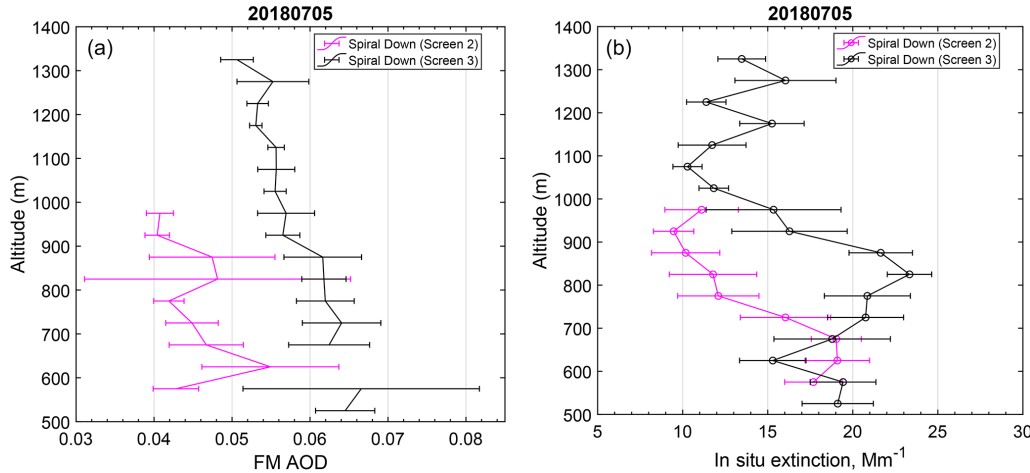

**Figure 10.** In-plume binned profile data obtained on 5 July during screens 2 and 3. The error bars represent standard deviations within each 50 m bin. **(a)** 4STAR FM AOD, **(b)** calculated in situ extinction coefficient at 500 nm. The in situ layer AOD between 575 and 975 m has increased by 0.002 ($\sim$ 33 %) from screen 2 to screen 3.

culated from the in situ extinction measurements. Based on these observations and the screen 1 to 3 increases in organic content and organic fraction, we attribute the observed AOD increase, at least in part, to the process of SOA formation.

Comparing out-of-plume 4STAR AODs to AERONET measurements shows that despite the slightly shorter atmospheric column 4STAR, AODs are marginally overestimated ($|\tau_{\text{diff}}| = 0.02$) with the difference increasing as a function of distance from the AERONET station. This suggests that either the 4STAR was still influenced by plumes with AODs higher than the AOSR background (presumably measured by AERONET) even at larger distances or that the AERONET AODs were generally underestimated during those flights (due for example to being influenced by cleaner air masses).

Finally, it is noted that low AOD AERONET measurements should be interpreted with care when they are used to represent the AOSR average. We showed that industrial plumes can be associated with significantly higher AODs in the vicinity of the emission sources, and their full extent is not captured by the ground-based instruments even with sources as close as 30–50 km. The impact of aerosol in-plume to the instantaneous change in solar irradiance is larger than the background aerosol for any other part of the screens and flight segments in the surrounding area, with changes occurring at the kilometer scales. By using the change in measured AOD and in situ aerosol optical properties (measured single scattering albedo and asymmetry parameter derived from Mie calculations of the aerosol size and refractive index), we calculated the change in instantaneous direct aerosol radiative effect (using the Fu–Liou radiative transfer code as part of LibRadtran; Emde et al., 2016). The aerosol within the industrial plume cooled the surface, through a decrease in the integrated net irradiance change (W m$^{-2}$), by an average reduction of incident light

by 25 %, as compared to the immediately surrounding background aerosol. This shows the potential to effect estimations of the aerosol's regional radiative impact, particularly if only using the AOD from the nearby AERONET site as sole indicator.

*Code availability.* The 4STAR raw data have been analyzed to produce AOD in this dataset using the open-source software published by the 4STAR Team (2018, https://doi.org/10.5281/zenodo.1492912). Other processing codes are available upon request.

*Data availability.* AERONET data are available at https://aeronet.gsfc.nasa.gov/ (NASA, 2021 TS6). The OSMC 4STAR data are available at https://doi.org/10.5281/zenodo.3517179. The OSMC in situ data are available at https://www.canada.ca/en/environment-climate-change/services/oil-sands-monitoring/monitoring-air-quality-alberta-oil-sands.html (Government of Canada, 2021).

*Supplement.* The supplement related to this article is available online at: https://doi.org/10.5194/acp-21-1-2021-supplement.

*Author contributions.* KB, MW, SL and JR conceived the study. MW, JL, JR and SL acquired the funding. MW and JR oversaw the use of 4STAR in Canada. KB and MW collected the 4STAR data on board the NRC Convair-580. KB, SL and KP analyzed the 4STAR direct-Sun measurements. KR, NTON and KB performed the SDA analysis. KP, SL and CF processed the sky scan data. TWC and KB analyzed nephelometer and CLAP data. MJW, KB and LN analyzed UHSAS and SP2 data. KB analyzed the AERONET data with contributions from KR, NTON and SL. RJ and SL provided engineering support for integrating and running the 4STAR on board

the NRC Convair-580. KB wrote the paper with contributions and reviews from all authors.

*Competing interests.* The authors declare that they have no conflict of interest.

*Acknowledgements.* We would like to acknowledge the NRC and ECCC project crew that supported the field project. We also acknowledge Warren Gore (retired) from NASA Ames Research Center for his efforts in bringing the 4STAR to Canada for the duration of the project. We thank Ihab Abboud and Vitali Fioletov for establishing and maintaining the Fort McMurray and Fort McKay AERONET sites used in this investigation.

*Financial support.* Funding for the NRC Convair-580 participation in the Oil Sands 2018 field project was provided by ECCC and NRC. This work was partially funded under the Oil Sands Monitoring (OSM) program but it is independent of any position of the OSM program. The US authors were supported in part by NASA Radiation Science Program, under the direction of Hal Maring.

*Review statement.* This paper was edited by Barbara Ervens and reviewed by two anonymous referees.

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

**Remarks from the typesetter**

TS1    Please note that I uploaded the new Supplement file.

TS2    Please give an explanation of why the figure needs to be changed. We have to ask the handling editor for approval. Thanks.

TS3    Please give an explanation of why this needs to be changed. We have to ask the handling editor for approval. Thanks.

TS4    Please give an explanation of why this needs to be changed. We have to ask the handling editor for approval. Thanks.

TS5    Please give an explanation of why the figure needs to be changed. We have to ask the handling editor for approval. Thanks.

TS6    Please confirm citation.

TS7    Please confirm creator.

TS8    Please provide date of last access and confirm the reference list entry for the data set in the data availability section.