# Peer review of "Airborne and ground-based measurements of aerosol optical depth of freshly emitted anthropogenic plumes in the Athabasca Oil Sands region"

_Atmospheric Chemistry and Physics, 2020_

## Referee Comment (RC1) · Anonymous Referee #1 · 7 Jan 2021

This paper reports airborne measurements of aerosol properties in the Athabasca area as part of the Oil Sands Measurement Campaign (OSMC) during 2018. Ground-based Sun photometry data collected in the area are also used. The manuscript is in scope for ACP. It is written clearly. The airborne data provide a good amount of detail on various aerosol plumes measured during the flights, which will be useful to the broader community, since (as the paper notes) they haven't been studied in as much detail as some other aerosol systems. One key point here is that the spatial scales of these plumes are such that they can be missed by the ground-based measurements. I would

have liked to see a bit more satellite imagery and possibly retrievals to provide a bit more context about spatial variation here. Additionally, the measurements revealed that the plumes had different size and absorption characteristics from one another, i.e. not all the plumes in the area are similar.

I don't have any big issues with the material presented here. I recommend publication following minor revisions. I would be happy to review the revised version, if the Editor would find it useful. My comments are as follows:

1. Throughout the paper, it was difficult to judge the scale of the area and the plumes. I suggest adding a scale in km to Figure 1 so the reader has a sense of size of the overall domain size.

2. I suggest a new figure (either one multiple panel, or one for each of the main flights discussed) be added to show a true-color image around the time of the flights? This would help the reader visually see what was going on. I am not sure if MODIS or VIIRS overflew around the right time (or maybe we will have got lucky and there's Landsat or Sentinel 2), but if not there are the new GOES sensors which are every 10 minutes or so. I looked on NASA Worldview for the days but wasn't sure if I could see the plumes – there were lots of clouds on some days too – so if the authors can provide the relevant imagery so we know what we are looking at, it would be helpful. Here is a link to June 9 imagery, not sure if the plume is visible here (there's a lot of cirrus too), or if the long url will make it through the ACP comment system unmangled: https://worldview.earthdata.nasa.gov/?v=-114.52655240204368,55.351947254962724,-108.18835611557154,58.463275379962724&t=2018-06-09-T21%3A46%3A08Z&l=Reference_Labels,Reference_Features,Coastlines,VIIRS_NOAA20_CorrectedReflectance_

3. Satellite retrievals of AOD would also be interesting to show, to reveal whether they resolved the plume structures or not. The MODIS Dark Target 3 km product could be useful here as it is finer than most others. Again, it's hard to know what the spatial scale is from the paper, so it's possible this would be too coarse already? And

if satellite products don't resolve the plume (either structurally or even as a hotspot) that is another interesting point (analogous to the AERONET spatial representation discussed by the authors for this area).

4. Page 5 line 29: the authors mention O'Neill et al (2016) as a reference for cloud screening based on the SDA (i.e. that the fine mode is unaffected). However Smirnov et al (2018) indicate that in the presence of cirrus (or dust) the SDA fine mode AOD can still be biased: https://www.sciencedirect.com/science/article/pii/S0022407317306131 Perhaps the authors can comment on this, particularly as there seemed to be some cloud cover in the satellite images on Worldview.

5. Page 6 lines 3-4: it might be covered in the references cited, but could the authors mention here whether the UHSAS size distribution retrieval requires assumptions about refractive index and if so how sensitive it is to that? This could be relevant as it is an optical sensor, and differences between plume refractive index could mask or magnify differences in particle size between plumes.

6. Any other caveats or relevant uncertainty sources associated with the in situ measurements should also be mentioned in Section 2.4.

7. Figure 3: Panel 3 shows UHSAS effective radii around 0.4 microns in Plume A. However, for the same flight (9 June), the lower panel of Figure S4 has all UHSAS data between 0.1 and 0.2 microns. Is this a plotting error in one of the figures, or am I misunderstanding what is shown?

8. Figure 5: do the authors believe the narrow peak in plume A around 0.42 microns is real, or could it be an instrumental/retrieval artefact? Any thoughts on what could cause this sharp feature?

9. Page 9 line 6: I am not sure it is quite right to say that AERONET sites are generally assumed to be representative of a distance 100 km around them. Most satellite retrievals use an averaging circle of order 25 km. Even for a model comparison, if it is at

1 degree, then the grid boxes are still only 110 km (i.e. a 55 km box if centered around the site) at the Equator and smaller at the poles. I understand the authors' point here but suggest revising the wording to not say "this distance is often taken as 100 km" because I don't believe that is true.

10. Figure 7: the caption notes that the horizontal bars on the AERONET panel here are standard deviation. What are the horizontal bars on the upper panel? This should be stated.

11. Figure 9: I think the caption should read 0.02 here, not 0.002, unless I am misunderstanding.

12. Figure S5: I do not think that the regression is valid here. The fitting, p value, and uncertainties are based on the assumption of independent draws from one population of data. What we have here is data from 4 separate flights. Each flight is likely to have some autocorrelation between observations from that flight, and it's not necessarily true that the difference vs. distance would be consistent between all flights. I suggest redrawing this to color code points from the individual points, and add a zero line but perhaps not a regression. The reader can draw their own conclusions and I'm not sure that the regression is needed for the understanding of the paper: I agree that there seems to be some relationship, but caution against over-interpretation based on a small sample of flights.

---

## Referee Comment (RC2) · Anonymous Referee #2 · 25 Jan 2021

Summary:

Overall, this is a well-written paper that presents a case study of two flights during an aircraft campaign in the Athabasca Oil Sands Region near Alberta, Canada. The focus of the paper is on how the 4STAR aerosol optical depth (AOD) observations on-board the aircraft compare with the ground-based AERONET observations at nearby sites. The aircraft observations are also compared with in situ aerosol measurements to provide additional context about the composition and size distributions of aerosols associated with individual pollution plumes. The campaign and data are clearly presented
and the conclusions seem sound. While the findings are not particularly surprising, this paper would be valuable to the community as an additional data point for interpreting how ground-based remote sensing observations of aerosol optical properties at specific sites compare to the variability associated with pollution plumes in the atmosphere, specifically in this case in the context of industrial pollution sources. I can recommend the paper for publication after some minor revisions and clarifications.

General Comments:

While this paper is presented as a specific case study, I wonder if it would be possible to comment more on the representativeness of the variability of spatial scale observed here. Since the focus was on comparing the aircraft AOD observations with the AERONET observations, it might be useful to understand more about how this compares with observations from previous aircraft campaigns. Are the spatial scales of the plumes observed during the OSMC campaign similar to what is typically observed by 4STAR?

I agree with Reviewer 1 that some additional context, such as satellite measurements, would be helpful for giving the reader a better overview of what is happening. Were there any lidar measurements on the flights that could help to provide context?

Specific Comments:

Line. 5 p. 2. "The fact that industrial plumes can be associated with significantly higher AODs in the vicinity of the emission sources than previously reported from AERONET can potentially have an effect on estimating the AOSR radiative impact." "Cursory radiative transfer calculations" indicating 25% increase over background were mentioned at the end of the paper. Could this be expanded upon? 25% increase in terms of what, W/m2 or AOD? This was not clear from the discussion on p. 13, lines 10-15. What were the assumptions going into the calculation here? Presumably this would be a smaller effect than 25% once it is averaged over the entire grid box that the AERONET observations of AOD might be used to estimate.

Figure 2. It might be useful to also show the variance on the average AOD values for each month over the 13 year period. That would be useful for understanding the context of the flight observations.

Figure 3. There are some points in the AOD time series in pane 1 that appear to potentially be artifacts during periods where there were changes in aircraft altitude (e.g. the very smooth lines between 15:42-15:44, 15:50-15:52, 15:58-16:00, and 16:08-16:10). This is also the case for the UHSAS fine mode observations in pane 2 – can you comment on whether these are interpolation artifacts (and if so remove this data from the plot) or whether there is some other reason (like differences in averaging time) that the observations during these periods are significantly smoother than during the horizontal legs of the flight observations? Figure 6 and Fig. S4 also show similarly smooth periods in some of the time series.

Figure 4. It might be nicer visually to plot so that the organic aerosol mass portion starts at the bottom of each bar. This would make it easier for the reader to directly compare the organic aerosol mass across altitude levels/plumes and see that it stays relatively constant.

Can you speculate about the origins of the June 9th flight plume A and plume B based on their composition?

p. 8 . For context, could you add more details about what this facility is? Is it an oil processing plant?

Is there any way to judge the vertical extent of plume A relative to plume B?

p.10. Was there any estimate of the contribution of the AOD below flight level for the 4STAR measurements?

Figure S5. Could you similarly show the relative comparison between the Fort McMurray and Ft. McKay AERONET observations? This might help support the point in the first paragraph on p. 11.

p. 11 – Can you comment on the relative time scales expected for the plume's AOD to increase because of SOA formation compared with the time scale for the plume's AOD to decrease due to plume dilution with the background? Also, can you compare with the SSA observations, as SSA would also tend to be correlated with SOA formation?

Figure 7. Was there variability in AOD for different times of the day for the AERONET observations? Were the AERONET observations at approximately the same time as the flight observations? Also, can you clarify if the time shown on the axis for Figures 3, 6, and S4 is local time or UTC?

Typos: P. 7 Line 29-31 – This is referencing Figure 2, but it should be Figure 1.
* * *

---

## Author Comment (AC1) · 30 Apr 2021

Responses to Reviewer 1

April 2021

Green – reviewer's comment
Black – authors' response

Changes to text were made only when explicitly stated

**Reviewer # 1**

This paper reports airborne measurements of aerosol properties in the Athabasca area as part of the Oil Sands Measurement Campaign (OSMC) during 2018. Ground-based Sun photometry data collected in the area are also used. The manuscript is in scope for ACP. It is written clearly. The airborne data provide a good amount of detail on various aerosol plumes measured during the flights, which will be useful to the broader community, since (as the paper notes) they haven't been studied in as much detail as some other aerosol systems. One key point here is that the spatial scales of these plumes are such that they can be missed by the ground-based measurements. I would have liked to see a bit more satellite imagery and possibly retrievals to provide a bit more context about spatial variation here. Additionally, the measurements revealed that the plumes had different size and absorption characteristics from one another, i.e. not all the plumes in the area are similar. I don't have any big issues with the material presented here. I recommend publication following minor revisions. I would be happy to review the revised version, if the Editor would find it useful.

We thank the reviewer for constructive comments and provide detailed responses below.

My comments are as follows:

1. Throughout the paper, it was difficult to judge the scale of the area and the plumes. I suggest adding a scale in km to Figure 1 so the reader has a sense of size of the overall domain size.

We added a scale to Figure 1.

2. I suggest a new figure (either one multiple panel, or one for each of the main flights discussed) be added to show a true-color image around the time of the flights? This would help the reader visually see what was going on. I am not sure if MODIS or VIIRS overflew around the right time (or maybe we will have got lucky and there's Landsat or Sentinel 2), but if not there are the new GOES sensors which are every 10 minutes or so. I looked on NASA Worldview for the days but wasn't sure if I could see the plumes – there were lots of clouds on some days too – so if the authors can provide the relevant imagery so we know what we are looking at, it would be helpful. Here is a link to June 9 imagery, not sure if the plume is visible here (there's a lot of cirrus too), or if the long url will make it through the ACP comment system unmangled: https://worldview.earthdata.nasa.gov/?v=- 114.52655240204368,55.351947254962724,- 108.18835611557154,58.463275379962724&t=2018- 06-09- T21%3A46%3A08Z&l=Reference_Labels,Reference_Features,Coastlines,VIIRS_NOAA20_Cor rectedReflectance_

We added a new figure to section 3.2, showing a singular case of MODIS AOD as compared to 4STAR AOD of the Oil Sands processing plume.

3. Satellite retrievals of AOD would also be interesting to show, to reveal whether they resolved the plume structures or not. The MODIS Dark Target 3 km product could be useful here as it is

finer than most others. Again, it's hard to know what the spatial scale is from the paper, so it's possible this would be too coarse already? And if satellite products don't resolve the plume (either structurally or even as a hotspot) that is another interesting point (analogous to the AERONET spatial representation discussed by the authors for this area).
The figure added in response to the above comment has another section showing the comparison of MODIS Dark Target AOD to the airborne sampling from 4STAR. However, it is hard to bring any judgement to the limited sampling presented here, and the potentially large offsets in time, and therefore plume evolution and advection. See the additional new paragraph in section 3.2.

4. Page 5 line 29: the authors mention O'Neill et al (2016) as a reference for cloud screening based on the SDA (i.e. that the fine mode is unaffected). However Smirnov et al (2018) indicate that in the presence of cirrus (or dust) the SDA fine mode AOD can still be biased: https://www.sciencedirect.com/science/article/pii/S0022407317306131 Perhaps the authors can comment on this, particularly as there seemed to be some cloud cover in the satellite images on Worldview.
Smirnov et al. (2018) (SM) is focused on a very particular application of the SDA (Arola et al., 2017 or AR) where AR attempted to retrieve FM AOD from CIMEL measurements that were known to be contaminated by cirrus cloud. Basically SM criticized AR for ignoring and/or assuming that the SDA was somehow exempt from the well-known forward scattering effect of cirrus clouds into the large FOV of CIMEL instruments*. The SDA-focus of SM's paper is also misleading: in actual fact, the FOV effect is not some kind of unintended consequence of applying the SDA (the FM AOD is underestimated as a result of all standard AERONET AOD products being underestimated).

* which, for the record, they did not do: AR stated clearly that "It is likely, however, that the fine-mode AOD is underestimated when cirrus ice crystal clouds overlay the aerosol, due to strong forward scattering into the field of view of the sun photometer (A. Smirnov, personal communication, 2016)."

We also note that dust would generally not be a problem unless the dust optical depths (DODs) are very large (which, with DODs < 0.05, was never the case): the FOV effects of thin cirrus on the standard AOD and the FM AOD are already small enough: the effect of significantly smaller dust particles of relatively weak DOD would be negligible

5. Page 6 lines 3-4: it might be covered in the references cited, but could the authors mention here whether the UHSAS size distribution retrieval requires assumptions about refractive index and if so how sensitive it is to that? This could be relevant as it is an optical sensor, and differences between plume refractive index could mask or magnify differences in particle size between plumes.
Yes, UHSAS sizing will be dependent on the particle refractive index. We added the following phrase to section 2.4:

"The UHSAS sizing was calibrated using NIST traceable polystyrene latex (PSL) nanospheres. Sizing of the UHSAS is dependent on the refractive index and shape of the particles. Differences in refractive index has been estimated to result in a 10% uncertainty in the sizing of the UHSAS (Kupc et al, 2018)."

6. Any other caveats or relevant uncertainty sources associated with the in situ measurements should also be mentioned in Section 2.4.
We added more details to Section 2.4 regarding the AMS uncertainties/method.

7. Figure 3: Panel 3 shows UHSAS effective radii around 0.4 microns in Plume A. However, for the same flight (9 June), the lower panel of Figure S4 has all UHSAS data between 0.1 and 0.2 microns. Is this a plotting error in one of the figures, or am I misunderstanding what is shown?
It's unclear what the reviewer is referring to. Figure 3 is plotted for the June 9 flight while Figure S4 refers to the June 18 flight. Regardless, panel 3 of Figure 3 shows maximum values of $r_{eff}$ below 0.2 (right-hand scale) which is similar Figure S4.

8. Figure 5: do the authors believe the narrow peak in plume A around 0.42 microns is real, or could it be an instrumental/retrieval artefact? Any thoughts on what could cause this sharp feature?
Upon further investigation we concluded that this peak was likely an instrument artifact as it was not supported by larger-size particle spectrometers (such as FSSP and FCD). It looks like the problematic bins are close to the boundary of two gain stages in UHSAS calibration curve which could be a potential explanation for this problem. We excluded the problematic 5 bins (r= 0.382 – 0.428 μm) from analysis and added the following sentence to the instrumentation section:

"For some flights we noticed abnormally high particle counts in 5 bins between the radius of 0.382 and 0.428 μm. This peak was not supported by other particle spectrometers on the aircraft and is likely an instrument artefact. We removed the problematic data from further analysis and suspect that the issue is related to the uncertainties in UHSAS calibration curve consisting of several individually chosen gains."

9. Page 9 line 6: I am not sure it is quite right to say that AERONET sites are generally assumed to be representative of a distance 100 km around them. Most satellite retrievals use an averaging circle of order 25 km. Even for a model comparison, if it is at 1 degree, then the grid boxes are still only 110 km (i.e. a 55 km box if centered around the site) at the Equator and smaller at the poles. I understand the authors' point here but suggest revising the wording to not say "this distance is often taken as 100 km" because I don't believe that is true.
We would tend to disagree with such a small circle of AOD "expansiveness" and suggest that the satellite retrievals are the more likely source of spatial AOD incoherence : the reviewer will recall that on page 11, line 1 of our paper we cite Sioris et al. (2017) whose AEROCAN-wide correlation coefficients required 500 km of interstation distance to drop off by 50%* (in fact the fine mode correlation length** of their Figure 4a was closer to 1000 km)

* with the Fort Mckay to Fort McMurray contribution to the correlation curve that produced that correlation length estimate being close to the AEROCAN-wide average (personal communication with Chris Sioris)
** which the reviewer will agree is a more fundamental indicator of the spatial influence of AOD than a satellite-based inference

10. Figure 7: the caption notes that the horizontal bars on the AERONET panel here are standard deviation. What are the horizontal bars on the upper panel? This should be stated.
The horizontal error-bars are the std values at each altitude level. The caption was modified to read:

"In both panels the horizontal error bars indicate the AOD standard deviation representing spatial variability of the plumes (per altitude std values for 4STAR and daily std values for AERONET)"

We are only talking about the difference in layer AOD between screens 2 and 3 (575-975 m - common altitude range between the two spirals), so 0.002 is the correct number.  It's the relative increase that's more important here.  The calculated AOD of the entire vertical column sampled by neph+CLAP (525-1325 m for screen 3) is actually 0.013 which was stated in the text.

The figure has been replotted to separate the contributions from each flight, and each regression line applied to the individual flight days. All the flight days show increasing divergence of the AOD with increasing distance from the AERONET site, albeit with different magnitude of slopes and p-values, which are also reported. See the amended figure in the supplement. The main text remains unchanged.

---

## Author Comment (AC2) · 30 Apr 2021

Responses to Reviewer 2

April 2021

Green – reviewer's comment
Black – authors' response

Changes to text were made only when explicitly stated

**Reviewer # 2**

Summary:

Overall, this is a well-written paper that presents a case study of two flights during an aircraft campaign in the Athabasca Oil Sands Region near Alberta, Canada. The focus of the paper is on how the 4STAR aerosol optical depth (AOD) observations on-board the aircraft compare with the ground-based AERONET observations at nearby sites. The aircraft observations are also compared with in situ aerosol measurements to provide additional context about the composition and size distributions of aerosols associated with individual pollution plumes. The campaign and data are clearly presented and the conclusions seem sound. While the findings are not particularly surprising, this paper would be valuable to the community as an additional data point for interpreting how ground-based remote sensing observations of aerosol optical properties at specific sites compare to the variability associated with pollution plumes in the atmosphere, specifically in this case in the context of industrial pollution sources. I can recommend the paper for publication after some minor revisions and clarifications.
We thank the reviewer for constructive comments and provide detailed responses below.

General Comments:

While this paper is presented as a specific case study, I wonder if it would be possible to comment more on the representativeness of the variability of spatial scale observed here. Since the focus was on comparing the aircraft AOD observations with the AERONET observations, it might be useful to understand more about how this compares with observations from previous aircraft campaigns. Are the spatial scales of the plumes observed during the OSMC campaign similar to what is typically observed by 4STAR?
The aerosol plumes similar to those observed during OSMC have seldom been quantified previously. We added an additional figure (Figure 6, comparisons with MODIS AOD) and an accompanying paragraph in section 3.2 to discuss the spatial scales of the observed plumes. Furthermore, we updated Figure S5 which shows 4STAR-AERONET biases as a function of distance from AERONET.

I agree with Reviewer 1 that some additional context, such as satellite measurements, would be helpful for giving the reader a better overview of what is happening. Were there any lidar measurements on the flights that could help to provide context?
We added a case comparison where there are clear indications of a plume from MODIS true color and Dark Target AOD retrievals. See updated section 3.2 and related descriptions. Unfortunately, there were no lidar measurement on board during the campaign.

Specific Comments:

Line. 5 p. 2. "The fact that industrial plumes can be associated with significantly higher AODs in the vicinity of the emission sources than previously reported from AERONET can potentially have an effect on estimating the AOSR radiative impact." "Cursory radiative transfer calculations" indicating 25% increase over background were mentioned at the end of the paper. Could this be expanded upon? 25% increase in terms of what, W/m2 or AOD? This was not clear from the discussion on p. 13, lines 10-15. What were the assumptions going into the calculation here? Presumably this would be a smaller effect than 25% once it is averaged over the entire grid box that the AERONET observations of AOD might be used to estimate.

We expanded the last paragraph to include some description of the cursory radiative transfer calculations. Please see amended paragraph. While it was the hope of the authors to include more details on the radiative transfer calculations, this is beyond the scope of the paper.

Figure 2. It might be useful to also show the variance on the average AOD values for each month over the 13 year period. That would be useful for understanding the context of the flight observations.

We added error bars to Figure 2, representing standard deviation of the monthly mean AODs.

Figure 3. There are some points in the AOD time series in pane 1 that appear to potentially be artifacts during periods where there were changes in aircraft altitude (e.g. the very smooth lines between 15:42-15:44, 15:50-15:52, 15:58-16:00, and 16:08-16:10). This is also the case for the UHSAS fine mode observations in pane 2 – can you comment on whether these are interpolation artifacts (and if so remove this data from the plot) or whether there is some other reason (like differences in averaging time) that the observations during these periods are significantly smoother than during the horizontal legs of the flight observations? Figure 6 and Fig. S4 also show similarly smooth periods in some of the time series.

During transformation flights such as the one on June 9, at the end of each leg the aircraft makes a turn to rejoin the track at a higher altitude.  Depending on the steepness of the roll and sun position, sun tracking might be challenging during a turn resulting in occasional lack of 4STAR measurements between the altitude levels.  We replotted Figures 3, 7 and S4 to ensure that the disjointed 4STAR data segments are not connected by a straight line.  Other time-series do not appear to suffer from the same issue.

Figure 4. It might be nicer visually to plot so that the organic aerosol mass portion starts at the bottom of each bar. This would make it easier for the reader to directly compare the organic aerosol mass across altitude levels/plumes and see that it stays relatively constant. Can you speculate about the origins of the June 9th flight plume A and plume B based on their composition?

We replotted Figure 4 with the organic fraction starting at the bottom.  We also expanded section 3.2.1 to include a comment on the potential origins of the observed plumes (likely SML/Suncor emissions from upgrading the bitumen and/or mining activities).

p. 8 . For context, could you add more details about what this facility is? Is it an oil processing plant? Is there any way to judge the vertical extent of plume A relative to plume B?

We added more details to section 3.2.1 about the potential origins of the observed plumes. Both plumes were still clearly distinguishable (from each other and from the background) in the in-situ measurements at each altitude level up to the highest altitude of 1750 ft.  We can't speculate on the plume vertical extent beyond that point.

p.10. Was there any estimate of the contribution of the AOD below flight level for the 4STAR measurements?

p.10. Was there any estimate of the contribution of the AOD below flight level for the 4STAR measurements?

No, given that the first flight leg usually started at 500 ft (150 m) above ground level, we expect that our AOD measurements captured most of the vertical extent of the plumes. Including the lowest altitude levels would likely result in slightly larger 4STAR AOD values which would further reinforce our finding about the 4STAR-AERONET differences.

Figure S5. Could you similarly show the relative comparison between the Fort McMurray and Ft. McKay AERONET observations? This might help support the point in the first paragraph on p. 11.

One of the main reasons of not including Fort McKay in our analysis is its limited data availability for June-July 2018 with < 3h of data available on June 18 and no data acquired on June 24 and July 5.

p. 11 – Can you comment on the relative time scales expected for the plume's AOD to increase because of SOA formation compared with the time scale for the plume's AOD to decrease due to plume dilution with the background? Also, can you compare with the SSA observations, as SSA would also tend to be correlated with SOA formation?

There is no easy to estimate relative time scales of these two competing processes (SOA formation and plume dilution). Even if we knew how fast mass was diluted, separate from SOA formation, one would still need to determine the impact on AOD of these processes. Since we see an increase in both AOD and SOA, SOA formation outcompetes dilution (and deposition), so the AOD increase is actually a lower limit to the true AOD increase from just SOA formation. In section 3.3 we do provide SOA formation rates of 1003±193 kg/hr and 443±45 kg/hr for June 9 and July 5, respectively.

Figure 7. Was there variability in AOD for different times of the day for the AERONET observations? Were the AERONET observations at approximately the same time as the flight observations? Also, can you clarify if the time shown on the axis for Figures 3, 6, and S4 is local time or UTC?

Yes, AERONET could have significant AOD variations throughout the day, as expected. In fact, the original Figure 7 shows AERONET AOD daily standard deviations for 4 different days. In addition, Figure S4 shows an example of AERONET FM AOD time series for June 18 (little variation on that particular date) and Figure 8 gives 4STAR and AERONET comparisons for AODs that were acquired within 10 minutes of each other.

We modified the captions to make explicit references to UTC time.

Typos: P. 7 Line 29-31 – This is referencing Figure 2, but it should be Figure 1.
Fixed